# Chromatin reorganization drives overexpression of a Btaf1 variant underpinning hematopoietic aging

Le Zong[1,4], Bongsoo Park[1,4], Yaqiang Cao [2], Fei Ma [3], Ferda Tekin-Turhan [1], Wakako Kuribayashi[1], Keji Zhao [2] & Isabel Beerman [1] ✉

Age-associated hematopoietic stem cell (HSC) dysfunction is accompanied by dramatic transcription changes, but it remains unclear whether specific transcripts could orchestrate these HSC aging phenotypes. Here, we perform epigenetic profiling in male mice to investigate the regulatory mechanisms underlying the HSC aging transcriptome and screen for potential aging driver genes. We identify a looping structure formed between part of the *Btaf1* gene and the whole *Ide* gene in old HSCs which is accompanied by overexpression of a shorter variant of *Btaf1* (*nBtaf1*). Mechanistically, elevated expression of *nBtaf1* drives the aging-associated overexpression of HSC and megakaryocyte progenitor (MkP) signature genes via regulating TBP binding at their promoters, which contributes to HSC expansion and elevated MkP production in aged mice. ShRNA-mediated knockdown of *nBtaf1* restores a younger HSC transcriptome and specifically represses aging-associated HSC expansion and elevated MkP production. In summary, our data provide high resolution analysis of a dysregulated HSC aging epigenome and reveal a *Btaf1* variant that drives HSC aging phenotypes in mice.

Aging of hematopoietic stem cells (HSC) is associated with dramatic changes in the transcriptome and functional decline, including reduced reconstitution potential, myeloid-biased differentiation, and clonal expansion with elevated risk of cancer transformation in humans[1–5]. The HSC aging transcriptome has been well characterized with various techniques ranging from early studies using microarray[3,6–8] to the latest iterations using single-cell RNA-seq[9–12]. Svendsen et al. recently compiled the available transcriptome data of aged mouse HSC and identified a list of HSC aging signature genes shared among different studies[13]. Although many studies have reported HSCs have changes in frequency after aging with unique functional outputs[14–18], little is known in regard to the existence of potential driver genes that confer the HSC aging transcriptome and specific aging phenotypes.

Epigenetic alterations are recognized as hallmarks of aging and are often closely correlated with transcription[19,20]. However, due to the limited number of stem cells in the adult hematopoietic system, many aspects of epigenetic alteration during HSC aging remain unclear. Most published studies have focused on examining changes in DNA methylation[9,21–23] and chromatin accessibility[13,24–26], which can be conducted with smaller numbers of cells. It has been shown that chromatin accessibility was increased after HSC aging[13,26], along with elevated transposable element expression[21]. Histone modification changes during HSC aging have also been investigated[21,26], but technical advances have made higher resolution mapping feasible[27], allowing for more robust screening for drivers of aging.

Three-dimensional (3D) chromatin organization is another layer of regulation that is closely related to epigenetic mechanisms. Aging studies have proposed reorganization of chromosome architecture contributes to overall dysfunction[20,28,29], and explicitly in muscle stem cell aging, chromatin reorganization has been reported to be

[1]Epigenetics and Stem Cell Unit, Translational Gerontology Branch, National Institute on Aging, NIH, Baltimore, MD, USA. [2]Laboratory of Epigenome Biology, Systems Biology Center, National Heart, Lung, and Blood Institute, NIH, Bethesda, MD, USA. [3]Laboratory of Molecular Biology and Immunology, National Institute on Aging, NIH, Baltimore, MD, USA. [4]These authors contributed equally: Le Zong, Bongsoo Park. ✉e-mail: isabel.beerman@nih.gov

associated with variations in transcription factor (TF) binding and aberrant gene expression[30]. In the blood system, chromatin organization of HSC and differentiated cells has been reported in young mice[31,32] and our recent study investigated the chromatin architecture change in lineage-specific bone marrow (BM) progenitor B cells during aging[33]. However, the detailed 3D chromatin organization during HSC aging has not yet been fully explored.

Here, we present a comprehensive profiling of genome-wide data with purified HSCs from young (3 to 4 months) and old (24 to 26 months) mice, including 3D chromatin organization (Hi-C), chromatin accessibility (ATAC-seq), histone modifications (H3K4me3, H3K27me3, H3K36me3 and H3K27ac), and transcriptome (RNA-seq). We demonstrate that old HSCs (OHSC) have dysregulated epigenomes that shape their aging transcriptomes. Importantly, from the integration of these datasets, we identify a *Btaf1* variant that is overexpressed in OHSCs. Knockdown of this transcript in OHSCs restores a youthful HSC transcriptome and improves OHSC function in transplantation assays. These data not only provide a resource library of epigenetic analysis on a uniform HSC immunophenotype, but define aging-associated transcripts for targeting to improve the potential of an aged stem cell compartment.

## Results

### The chromatin in OHSCs is more open

Changes in epigenetic regulations of HSCs during aging are hypothesized to enforce altered functional potential. To investigate how epigenetic information changes during aging and to screen for potential aging driver genes, we performed integrated epigenetic profiling, including ATAC-seq, histone ChIP-seq and Hi-C, with mRNA-seq transcriptional profiling of HSCs (LSKCD34⁻Flk2⁻CD150⁺) purified from young and old mice. ATAC-seq was used to define age-associated changes in chromatin accessibility. Both PCA and unsupervised hierarchical clustering confirmed young and old HSC samples separated by age (Supplementary Fig. 1a, b). As an example, the *Mfsd1* locus is presented, showing a new peak at the end of the gene body in all eight OHSC samples, while the peaks at the gene promoter were similar in all young HSC (YHSC) and OHSC samples (Supplementary Fig. 1c). Differentially accessible regions (DAR) analysis (FC > 1.5, FDR < 0.01) identified more chromatin locations with increased accessibility (3827) rather than decreased accessibility (914) in OHSCs (Fig. 1a, b and Supplementary Data 1), indicating the chromatin in OHSCs is more accessible to binding factors. TF analysis of DARs showed enrichment for binding motifs of Batf, Runx1, Gata3, Ews:Fli1, and Meis1 in more accessible regions, while locations that lost accessibility were enriched for Spi1 and SpiB binding motifs (Supplementary Fig. 1d). The expression of *Batf* (a TF associated with gain-of-accessibility DARs) was increased, whereas the expression of *Spi1* and *SpiB* (TFs associated with loss-of-accessibility DARs) was reduced (Supplementary Fig. 1e). Histone modification evaluation at Batf-related DARs revealed sites that gained accessibility had higher levels of H3K4me3, lower levels of H3K27me3, and a marked increase in H3K27ac (Supplementary Fig. 1f, upper panel). Spi1-related DARs with reduced accessibility showed opposite changes, with decreased H3K4me3, increased H3K27me3, and a slight decrease of H3K27ac (Supplementary Fig. 1f, lower panel). Together, these data indicate that chromatin in OHSCs is more open, which may in turn enable altered TF binding patterns and changes in histone modification profiles.

### Increased chromatin accessibility in OHSCs correlates with increased enhancer activity and elevated transposable elements (TE) expression

As ATAC-seq peaks typically overlap with cis-regulatory elements, such as promoters and enhancers[34], the increased chromatin accessibility in OHSCs indicates there may be changes of enhancer activity during HSC aging. Comparisons of DARs and H3K27ac peaks showed about half of the DARs overlapped with H3K27ac peaks (Fig. 1c, d), and

increased chromatin accessibility was accompanied with elevated levels of H3K27ac (Fig. 1e). Using IDEAS, a software that integrates chromatin features to predict epigenetic states[35,36], we found the most dominant feature that increases in OHSCs was potential enhancer, featured by relatively high levels of H3K27ac and modest levels of H3K4me3 (Fig. 1f); most predicted genomic features were decreased in OHSCs, with regions of active transcription most robustly reduced (Fig. 1f). This is largely consistent with the global changes we observed in histone modifications, with a decrease of H3K4me3 and H3K27me3 at gene promoters and a global decrease of H3K36me3 at gene bodies in OHSCs (Fig. 1g). No change of H3K27ac was seen at promoter regions, but a modest increase of H3K27ac was seen at non-promoter regions (Fig. 1g and Supplementary Fig. 1g). Direct examination of changes in H3K27ac with fold change > 1.5 identified more increased peaks (3602) than decreased peaks (898) in OHSCs (Fig. 1h). Together, the increased contact frequency within TADs and the increased H3K27ac at non-promoter regions, support IDEAS prediction of elevated enhancer activity in OHSCs.

Analysis of DAR locations relative to TSS revealed that DARs were mostly located in regions 50–500 kb from TSS, with few sites located within 5 kb of the TSS (Supplementary Fig. 1h). Further analysis showed the majority of DARs were located at intergenic or intronic regions (Fig. 1i). As noncoding DNA regions, including intergenic and intronic locations, are enriched for transposable elements, we then examined whether TE expression was affected in OHSCs. A bias towards increased expression of TEs was seen, though there were also downregulated TEs in OHSCs (Fig. 1j). Analysis of the chromatin accessibility at these TEs found the chromatin was indeed more open at upregulated TEs and more closed at downregulated TEs (Fig. 1k and Supplementary Fig. 1i), supporting a connection between increased chromatin accessibility, differential enhancer activity, and TE expression.

### Age-associated histone modification changes are closely correlated with the HSC aging transcriptome

As histone modifications are closely correlated with transcription regulation, we next examined the correlation of our histone modification data to the HSC aging transcriptome. High correlation between replicates was observed for all antibodies and within age groups (Supplementary Fig. 2a). At the *Selp* locus, which has previously been reported to have increased transcription levels in OHSCs, we observe increased H3K4me3 and H3K27ac at the promoter, increased H3K36me3 in the gene body, and depletion of H3K27me3 across the gene (Supplementary Fig. 2b).

To examine whether histone alterations are associated with changes in mRNA expression, HSC-expressed genes were divided into three groups according to the expression levels: increased with age, decreased with age, or expressed but with no significant difference with age (Fig. 2a and Supplementary Data 2). We included the last comparison to evaluate if the corresponding histone modifications at these genes showed the same pattern of reduction as defined in global analysis in Fig. 1g. For genes downregulated in OHSCs, we observed a decrease of H3K4me3 at gene promoters, together with a large decrease of H3K36me3 in gene bodies. An increase of H3K27me3 was seen at these loci even though there is a global decrease in this mark (Supplementary Fig. 2c). A decrease in H3K27ac was also observed at the promoters of downregulated genes (Supplementary Fig. 2c). Interestingly, for genes with maintained expression, we still observed decreased H3K4me3, H3K27me3 and H3K36me3, with no change in H3K27ac (Fig. 2b and Supplementary Fig. 2d), similar to what was seen globally (Fig. 1g). For genes encoding RNAs more highly expressed in OHSCs, the levels of H3K4me3 and H3K36me3 in OHSCs were similar to, but not increased above, the levels seen in YHSCs, while H3K27ac showed modest increases and H3K27me3 was much lower in OHSCs (Supplementary Fig. 2e). Together these data suggest these age-associated histone marks largely correspond to altered mRNA expression in the OHSCs.

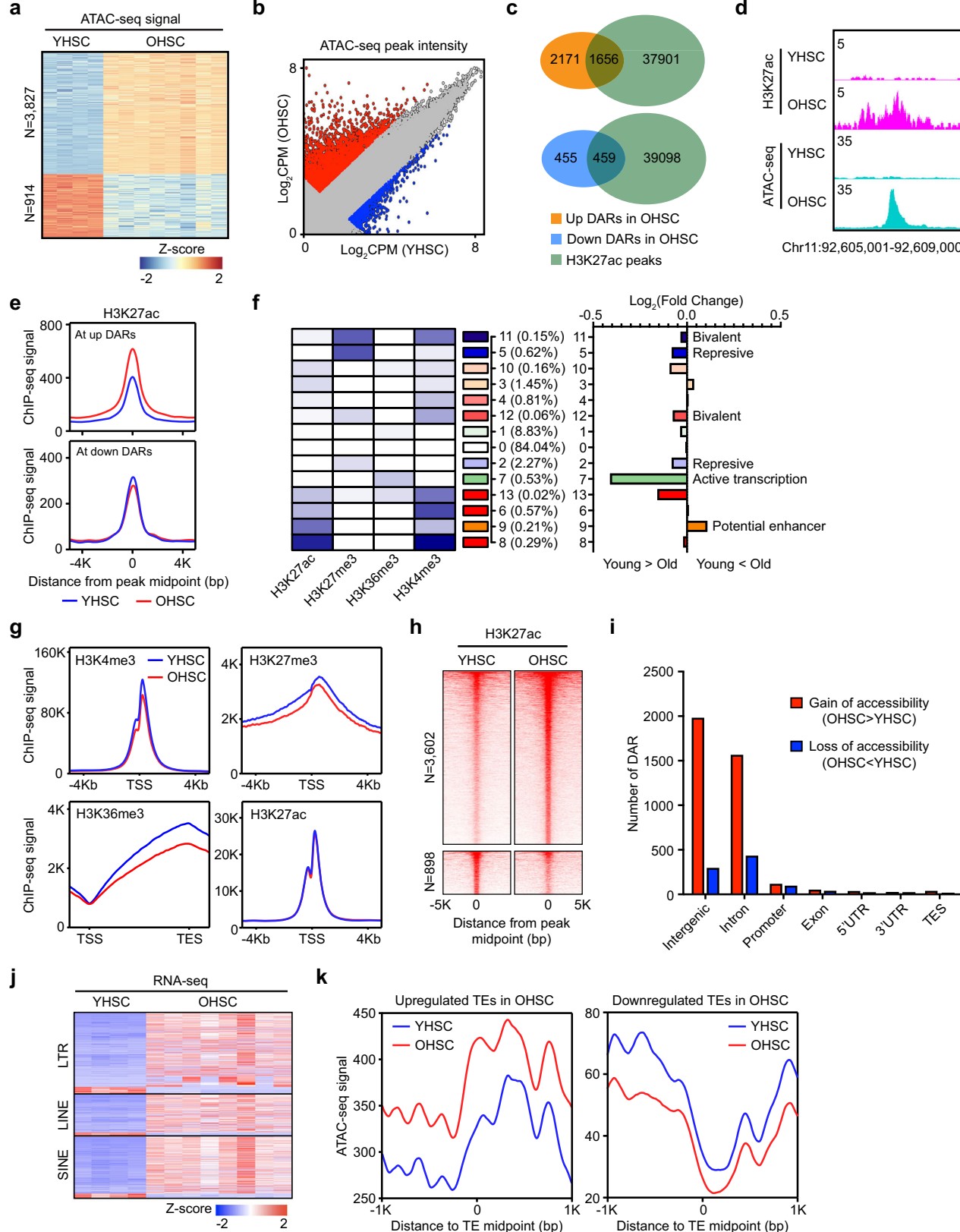

## Altered promoter bivalency contributes to dysregulated gene expression in OHSCs

Promoter bivalency, the occupancy of both permissive and repressive histone marks, is reportedly a common feature of stem cells and has been suggested to play important regulatory roles[9,21,27,37,38]. We wanted to establish bivalency in HSCs and determine if the promoter bivalency may be altered during aging, which could affect the priming and functional potential of OHSCs. To accurately assign bivalent domains, H3K27me3 peaks were first defined and assigned to a promoter only if the signal overlapped with or was ≤ 2 kb downstream of a transcription start site (TSS). This ensures H3K27me3 signals at intergenic regions upstream of the gene's promoter are not incorrectly assigned

**Fig. 1 | Increased chromatin accessibility in OHSCs correlates with elevated enhancer activity and transposable elements (TE) expression. a** Heatmap of differentially accessible regions (DAR) between young (*n* = 4) and old (*n* = 8) HSCs (FC > 1.5, FDR < 0.01). **b** Comparison of chromatin accessibility between young and old HSCs. Red dot: chromatin accessibility higher in OHSCs; blue dot: chromatin accessibility is lower in OHSCs. **c** Venn diagram showing the overlap between DARs and H3K27ac peaks. **d** IGV tracks showing an example region where H3K27ac peak and ATAC-seq DAR overlap. **e** Composite plots of H3K27ac signal at DARs that overlaps with H3K27ac peaks. **f** IDEAS chromatin state analysis and summary of fold change between young and old HSCs. Left panel: predicted chromatin states of the

mouse genome assigned according to the levels of histone modifications; the proportion of each state is shown in brackets; right panel: log-transformed change of each predicted chromatin state in OHSCs compared to YHSCs. **g** Composite plots of histone modification levels at all genes. TSS, transcription start site. TES, transcription end site. **h** Heatmap of changed H3K27ac peaks between young and old HSCs (FC > 1.5). **i** The number of DARs at different types of genomic locations. **j** Heatmap of differentially expressed TEs between YHSCs (*n* = 4) and OHSCs (*n* = 8) (FC > 1.5, FDR < 0.05). **k** Composite plots showing chromatin accessibility at up (left panel) or down (right panel) regulated TEs.

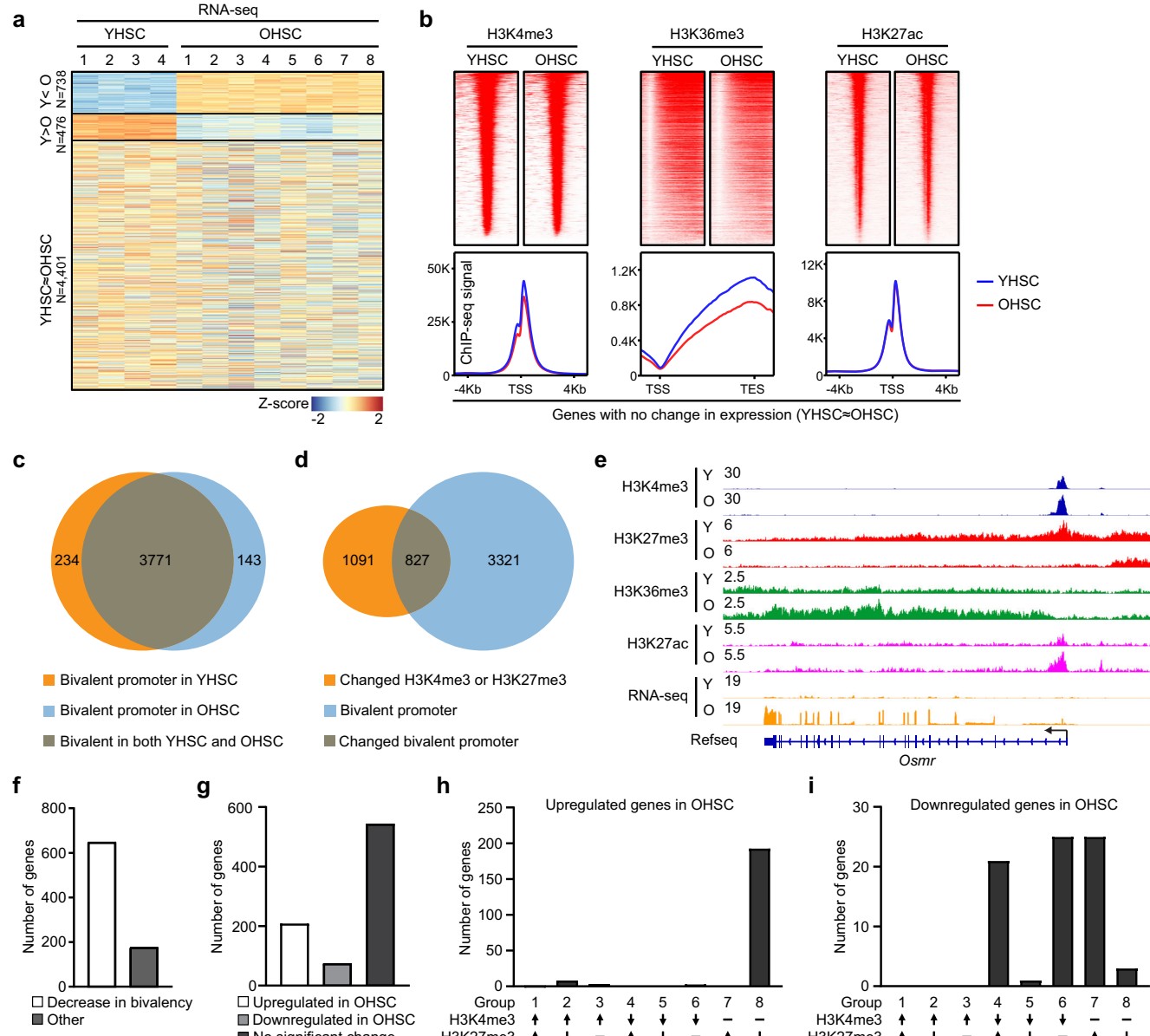

**Fig. 2 | Dysregulated histone modifications in OHSCs contribute to the HSC aging transcriptome. a** Heatmap of genes divided according to changes in expression. YHSC (*n* = 4), OHSC (*n* = 8). YHSC < OHSC or YHSC > OHSC (FC > 1.5 and FDR < 0.05) are classified as upregulated or downregulated; YHSC ≈ OHSC (FC < 1.05) category defined as similar expression. **b** Heatmap and composite plots of histone modifications generated with genes whose expression level was similar in young and old HSCs (FC < 1.05). **c** Venn diagram displaying the number of genes with bivalent promoters in young and old HSCs. **d** Venn diagram displaying the

number of bivalent promoters with changed H3K4me3 and/or H3K27me3 levels (FC > 1.5). **e** Histone modification and gene expression tracks at the *Osmr* locus. **f** Number of genes with decrease in bivalency at the promoter in OHSC (group 2, 4, 5, 6, and 8) and all other types of changes. **g** Number of genes with expression changes after a promoter bivalency change (FC > 1.5, FDR < 0.05). No significant change group includes genes not expressed, as well as genes that are expressed but with no significant age-associated change. Distribution of up (**h**) or down (**i**) regulated genes in OHSCs associated with changed promoter bivalency.

to bivalent domains. Four thousand five and 3914 bivalents promoters were identified in YHSCs and OHSCs, respectively, with 234 unique to YHSCs, 143 exclusive to OHSCs, and 3771 shared by both (Fig. 2c and Supplementary Data 3). Comparisons of these HSC-related bivalent regions do show overlap with embryonic stem cell bivalent promoters (~60%) but define over 1500 unique locations in adult HSCs (Supplementary Fig. 3a). Among the bivalent promoters, 827 showed altered H3K4me3 and/or H3K27me3 with aging (Fig. 2d).

Pathway analysis showed young HSC-specific bivalent promoters were enriched for pathways related to cell junction organization, whereas old HSC-specific bivalent promoters were enriched for cell division-related processes (Supplementary Fig. 3b, left panels). Bivalent promoters shared between young and old HSCs, as well as those changing with age, were enriched for cell junction organization and synaptic signaling pathways (Supplementary Fig. 3b, right panels).

When comparing HSC- and ESC-associated bivalent promoters, we found that HSC-specific promoters were enriched for embryonic development pathways, while shared promoters were enriched for cell junction organization and synaptic signaling pathways (Supplementary Fig. 3c, top panels). By contrast, ESC-specific bivalent promoters were enriched for MAPK cascade-related pathways, which may restrain ESC differentiation (Supplementary Fig. 3c, bottom panel). Notably, ESC-specific bivalent promoters were also enriched for blood cell differentiation pathways, which were not observed among HSC-related promoters (Supplementary Fig. 3c, left panels). This suggests that many blood lineage-associated promoters are bivalent in ESCs—allowing rapid activation during lineage commitment—but lose bivalency and become activated upon differentiation into HSCs. Interestingly, many HSC bivalent promoters remain linked to embryonic development and synaptic signaling (Supplementary Fig. 3c, top left panel), indicating a poised state that may reflect the plasticity of HSCs and their potential to revert toward earlier developmental programs.

To determine whether any of the changes in bivalency might lead to altered stem cell function, we looked for bivalency changes associated with transcription changes. We found such a change at *Osmr*, a gene which has been shown to be involved in the maintenance of erythroid and megakaryocyte progenitors[39]. In OHSCs, there was a depletion of H3K27me3 at the *Osmr* promoter and across the gene body compared to the YHSCs; meanwhile, the levels of H3K4me3 at the promoter and H3K36me3 in the gene body were increased (Fig. 2e). Together, these changes led to elevated expression of *Osmr* in OHSCs and may contribute to the altered lineage potential of the aged HSCs. Meanwhile, some bivalent promoters are preserved during HSC aging (Supplementary Fig. 4a).

To more clearly understand the bivalency changes, we checked all combinations of H3K4me3 and H3K27me3 changes at bivalent promoters, and 8 groups were defined (Supplementary Fig. 4b). The majority of changed bivalent promoters associated with aging in HSCs showed decrease in bivalency (Fig. 2f) and many were associated with changes in gene expression (Fig. 2g). Distribution analysis of genes with changed expression showed almost all upregulated genes were from group 8, suggesting that loss of H3K27me3 was the main contributor to the elevated expression of genes with bivalent promoters (Fig. 2h). Downregulated genes were mainly from groups 4, 6, and 7, associated with loss of the active mark H3K4me3, increase of the repressive mark H3K27me3 or both (Fig. 2i). Together, these data support altered promoter bivalency contributes to dysregulated gene expression in OHSCs. While we were able to identify a few specific genes, such as *Osmr*, which may contribute to altered priming of HSC differentiation, the transcripts with altered bivalency were not significantly enriched for pathways involved in HSC lineage commitment or stemness.

## Dysregulation of chromatin organization in OHSCs features decreased long-range and increased short-range interactions of heterochromatin

To expand upon the epigenetic alterations associated with aging, we evaluated changes in large-scale chromatin organization by generating Hi-C data from young and old HSCs. Initial quality control analysis showed high levels of similarity between biological replicates (Supplementary Fig. 5a and Supplementary Data 4) and replicates from the same age group cluster together (Supplementary Fig. 5b). Resolution estimation plots demonstrate that the Hi-C data are well powered for analyses at ≥25 kb resolution, which is sufficient for reliable compartment and TAD identification (Supplementary Fig. 5c). Closer analysis of chromatin interactions examining Hi-C contact matrices of all chromosomes showed no uniform structural alterations between same-aged samples (Supplementary Fig. 5d, top two panels), indicating high similarity between replicates; in contrast, comparisons between young and old HSCs revealed many changes (Supplementary Fig. 5d, last three rows), indicating the chromatin organization was altered in OHSCs. Examination of chromosomal subregions further confirmed age-associated changes in chromatin organization in OHSCs (Supplementary Fig. 5e).

We interrogated how 3D chromatin changed in OHSCs by examining the frequency distribution of intra-chromosomal contacts. This revealed decreased long-range (>5 Mb) and increased short-range (<5 Mb) interactions in OHSCs (Fig. 3a). More explicitly, Juicebox visualization[40] of interactions on representative chromosome 19 in YHSCs and OHSCs showed general similarity in overall patterns (Fig. 3b, left two panels), but overlap of the comparison between young and old HSC interactions (OHSC-YHSC) showed increased short-range interactions (red signal near diagonal) and decreased long-range interactions in OHSCs (blue signal away from diagonal) (Fig. 3b, right panel). To establish whether these changes in interactions occurred at distinct spatial regions containing active or inactive chromatin, we investigated the chromatin interactions at the compartment level (A and B), in which the A compartment typically correlates with regions of euchromatin, and the B compartment has more heterochromatin features[41]. Interestingly, decreased long-range interactions were enriched between B compartments as visualized by the blue clusters away from the axis in Fig. 3b (right panel). An example of a B compartment - a subregion of chr19 shown in Fig. 3b—exhibits low gene density and transcription, with histone profiles suggestive of heterochromatin (Fig. 3c). To test if this reduction in B–B compartment contact was a general phenomenon across the whole genome, contact frequency between all compartments was analyzed (A–A, A–B, and B–B), and only interactions between B compartments (B–B) showed a significant decrease in OHSCs (Fig. 3d). Among the top 200 decreased compartment interactions, the majority (193 out of 200) were B–B compartment interactions (Fig. 3e), further supporting that the decreased long-range interactions in OHSCs were mainly interactions between B compartments.

B compartments largely overlap with lamina-associated domains (LAD) as heterochromatin is often found closer to the nuclear lamina versus the nucleus core[42,43]. Thus, the loss of B–B compartment interactions implied that the organization of lamina associated chromatin might be altered. Of the 3 lamin-coding genes, only the expression of *Lmna* (encoding lamins A and C) was significantly decreased in OHSCs, while the expression of *Lmnb1* and *Lmnb2* (encoding lamin B1 and lamin B2) was not changed (Supplementary Fig. 5f). Fittingly, lamin A/C has been shown to be involved in the regulation of LAD and A/B compartment organization[43] and chromatin architecture changes during the HSC aging process[44]. Thus, the reduced *Lmna* expression and decreased contact frequency between B compartments suggest a loss of spatial regulation of chromatin architecture in OHSCs compared to YHSCs.

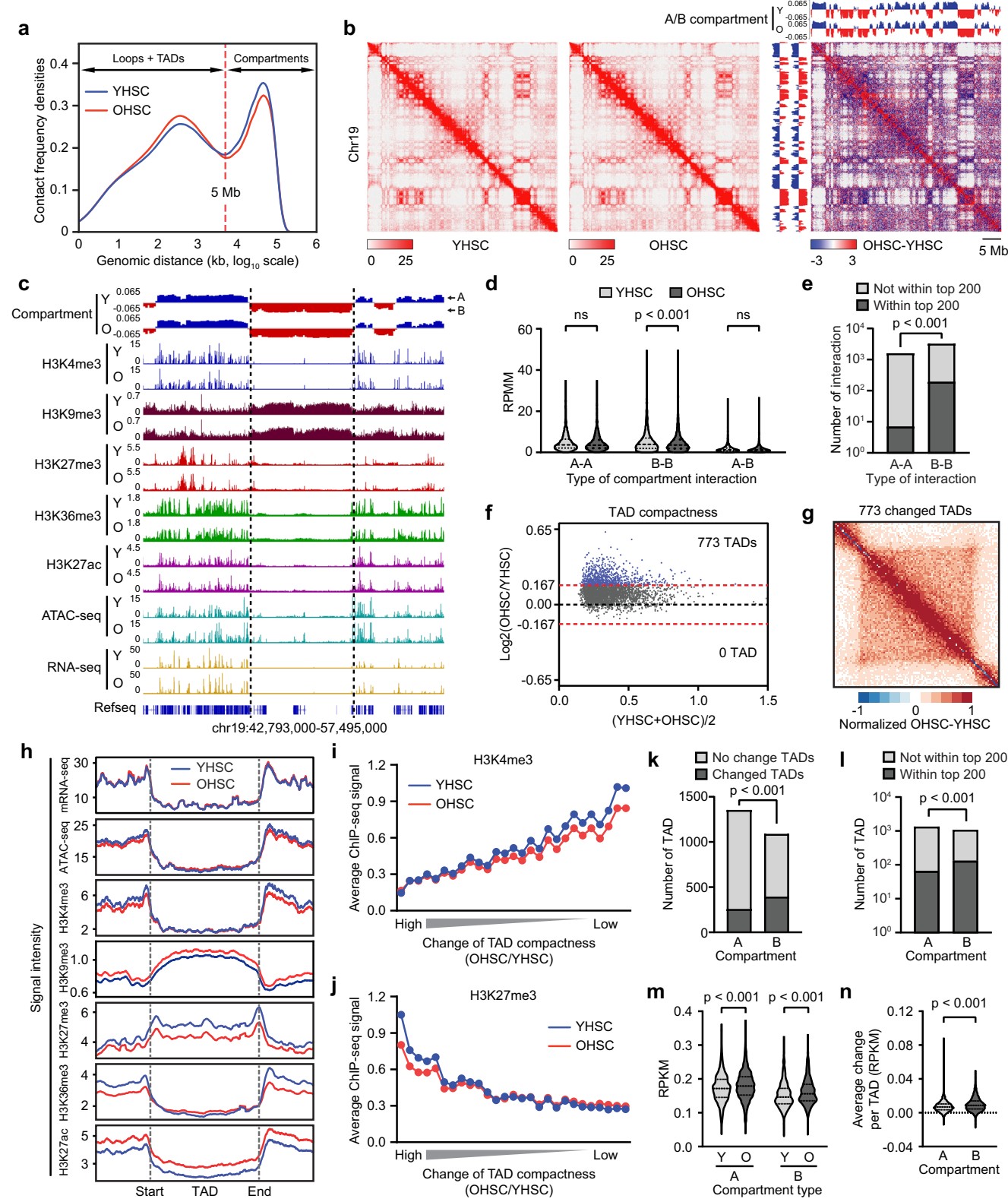

We next analyzed age-associated changes of shorter-range interactions within topologically associating domains (TAD). TADs are sub-Mb structural features with distinct boundaries that have increased interactions within themselves due to physical proximity[45]. To establish a cutoff for TADs with altered compactness, we compared the replicates of YHSCs and OHSCs, using the largest difference between replicates as the threshold to define a changed TAD (Supplementary Fig. 6a). Comparisons between young and old HSCs showed a general trend of increased TAD compactness in OHSCs, with 773 TADs above the cutoff (Fig. 3f). An aggregation plot comparing intensity between YHSCs and OHSCs of all changed TADs showed a clear TAD pattern (box shape) with increased intensity in OHSCs (Fig. 3g). As TADs are sub-Mb structures, these results are consistent with the observed increase of short-range interactions in OHSCs (Fig. 3a). Analysis of the chromatin state at these TADs with increased interaction in OHSCs showed depletion of active chromatin features (compared to regions

**Fig. 3 | Decreased long-range and increased short-range interactions of heterochromatin in OHSCs. a** Frequency distribution of intrachromosomal contacts generated with young and old HSC Hi-C data. TAD, topologically associating domain. **b** Hi-C contact matrices of chromosome 19, plotted at 100 kb resolution with Juicebox[40]. **c** Expanded integrative genomics viewer (IGV)[78] view of the chromatin state at a representative B compartment and its surrounding regions. **d** Contact frequency between the same (A–A, B–B) and different (A–B) compartments. Compartments (size > 1 M) shared by young and old HSCs were used. RPMM, reads per million base pairs per million reads. ns, not significant. Data are shown as violin plots. Dotted lines indicate the median and interquartile range. Two-tailed t-test. Source data are provided as a Source data file. **e** Number of A–A or B–B compartment interactions within, or outside of, the top 200 decreased compartment interactions in OHSCs. Two-sided Fisher's exact test. **f** Comparison of

TAD compactness between young and old HSCs showing 773 TADs with increased compactness in OHSCs. **g** Aggregation plot showing intensity changes of the 773 TADs with increased compactness in OHSCs. **h** Composite plots of the chromatin state at the 773 age-increased TADs. H3K4me3 (**i**) and H3K27me3 (**j**) signal at TADs (bin size = 100 TADs) sorted by change of compactness (OHSC/YHSC) from high to low. **k** Number of changed and unchanged TADs within A or B compartments. Two-sided Fisher's exact test. **l** Number of TADs assigned to A or B compartments within, or outside of, the top 200 increased TADs in OHSCs. Two-sided Fisher's exact test. **m** Average intensity of TADs from A or B compartments. Data are shown as violin plots. Dotted lines indicate the median and interquartile range. Two-tailed t-test. Source data are provided as a Source data file. **n** Average change per TAD in A and B compartments. Data are shown as violin plots. Dotted lines indicate the median and interquartile range. Two-tailed t-test. Source data are provided as a Source data file.

outside the TADs)—i.e., accessible chromatin, gene expression and active histone marks (H3K4me3, H3K36me3 and H3K27ac)—but contained repressive marks, H3K9me3 and H3K27me3 (Fig. 3h). Intriguingly, within these TADs, there appears to be lower levels of H3K27me3 and higher levels of H3K27ac in the OHSCs compared to the YHSCs (Fig. 3h and Supplementary Fig. 6b). To further validate this change in histone marks within TADs, all TADs were ranked by degree of change in compactness (OHSC/YHSC) from high (more compact in OHSCs) to low, and chromatin state of bins (groups of 100 TADs) was established. Consistent with the previous result (Fig. 3h), TADs with the largest change of compactness showed more repressive features (H3K9me3 and H3K27me3) and lack of permissive marks (Fig. 3i, j, and Supplementary Fig. 6c). As the change of compactness became smaller, TADs started to show more permissive features (Fig. 3i, j, and Supplementary Fig. 6c).

As the domains with increased repressive marks and loss of permissive marks again may indicate many changes of chromatin interaction occur in heterochromatin, we checked whether TADs with increased contact frequencies were mainly from B compartments. There were slightly lower numbers of overall TADs within B compartments than within A compartments, however there is a significant enrichment for TADs with increased contact frequency in B compartments (Fig. 3k). Moreover, for the top 200 TADs with increased compactness in OHSCs, the ratio of TADs within B compartments was over 2-fold higher compared with the ratio of TADs within A compartments (Fig. 3l). The contact frequency of TADs within both A and B compartments was increased in OHSCs (Fig. 3m) and was associated with slightly elevated levels of H3K27ac (Fig. 3h), but the average increase in contact frequency per TAD was higher for those within the inactive B compartments than TADs within more active A compartments (Fig. 3n). In summary, Hi-C results show the chromatin structure of HSC is altered during aging, leading to decreased long-range contact between B–B compartments and increased short-range interactions in both active and repressed compartments, with an enrichment in B compartments, in OHSCs.

### Elevated expression of a short *Btaf1* variant in OHSCs

As the goal of our analysis was to define potential driver transcripts of aging phenotypes, we analyzed the composite epigenomic libraries to identify such candidates. Starting from the changes in chromatin organization, we determined that among the top 5 TADs with increased contact frequency in OHSCs, the Chr19 TAD encompassed differentially expressed genes (DEG) defined from our young and old HSCs RNA-seq analysis (Supplementary Fig. 7a and Supplementary Data 5). Within this TAD, we identified an enriched chromatin interaction specific to OHSCs (Fig. 4a, upper panel). This interaction was formed by part of *Btaf1* and the whole *Ide* gene (Fig. 4a, middle panel). Interestingly, the formation of this looping structure was accompanied by elevated expression of a *Btaf1* variant (*nBatf1*), which shares the promoter and first 13 exons with canonical *Btaf1* but has its own 3'UTR defined by mRNA sequence analysis (Fig. 4a, lower panel). Histone

modification profiles also support increased expression of *nBtaf1*, as the levels of the active transcription-related mark H3K36me3 were increased exclusively within the range of this variant, together with increased H3K4me3 and H3K27ac at the promoter (Fig. 4a, lower panel). Quantification of the shorter and canonical *Btaf1* variants from total mRNA sequence data showed only expression of *nBtaf1* was increased in OHSCs, while the expression of the canonical *Btaf1* was not changed (Fig. 4b). *Ide*, the other gene associated with this loop, also had increased expression in OHSCs, together with increased active histone marks (Supplementary Fig. 7b, c).

### Knockdown of the shorter *Btaf1* variant leads to reduced HSC expansion and decreased production of megakaryocyte progenitors

As *Btaf1* encodes a TATA box-binding protein-associated factor (TAF) important for regulating the transcription initiation of genes by RNA polymerase II (Pol II)[46], the elevated expression of *nBtaf1* in OHSCs may play a large role in altered transcription contributing to aging phenotypes. To test this hypothesis, we designed shRNA to specifically target the expression of the shorter *Btaf1* variant. As *nBtaf1* shares the promoter and exons with canonical *Btaf1*, we targeted the unique 3'UTR. We verified that lentiviral expression of two independent shRNAs knocked down expression of the *nBtaf1* but did not alter expression of canonical *Btaf1* (Supplementary Fig. 7d–f).

We then performed a transplantation assay with control (scramble shRNA) and *nBtaf1* knockdown OHSCs to examine if reduced levels of *nBtaf1* could mitigate aging phenotypes (Fig. 4c). Peripheral blood analysis of recipient mice showed no change of total donor chimerism, GFP+ frequency (donor cells with shRNA), or lineage outputs (Supplementary Fig. 8a–d), indicating the elevated expression of *nBtaf1* in OHSCs does not affect the production of peripheral blood myeloid and lymphoid cells. Whole BM analysis over 4 months post-transplant also showed no change in total donor chimerism or GFP+ frequency from the knockdown OHSCs (Supplementary Fig. 8e, f). However, we found the frequency of megakaryocyte progenitors (MkP) was significantly reduced from *nBtaf1* knockdown donor OHSCs, but not changed in donor-derived GFP- cells (donor cells without knockdown) (Fig. 4d, e). The decrease in MkPs production after *nBtaf1* knockdown is similar to the lower frequency of MkPs seen in young animals compared to old (Fig. 4f).

To establish whether the knockdown of *nBatf1* affects the more primitive populations, we examined the frequency of hematopoietic stem and progenitor cells (Supplementary Fig. 8g) and found the HSC and HPC-2/MPP2 populations were decreased after *nBtaf1* knockdown (Fig. 4g, h), while the frequencies of HPC-1/MPP3-4 and MPP/MPP5 were not changed (Supplementary Fig. 8h, i). Consistent with MkP decrease, HSC and HPC-2/MPP2 preferentially differentiate to megakaryocyte progenitors, while HPC-1/MPP3,4 mostly differentiate to CMP, GMP, and lymphoid cells[47–49]. We next examined *nBtaf1* expression and chromatin state across myeloid progenitors (CMP, GMP, and MEP). RNA-seq analysis revealed the highest *nBtaf1* expression in

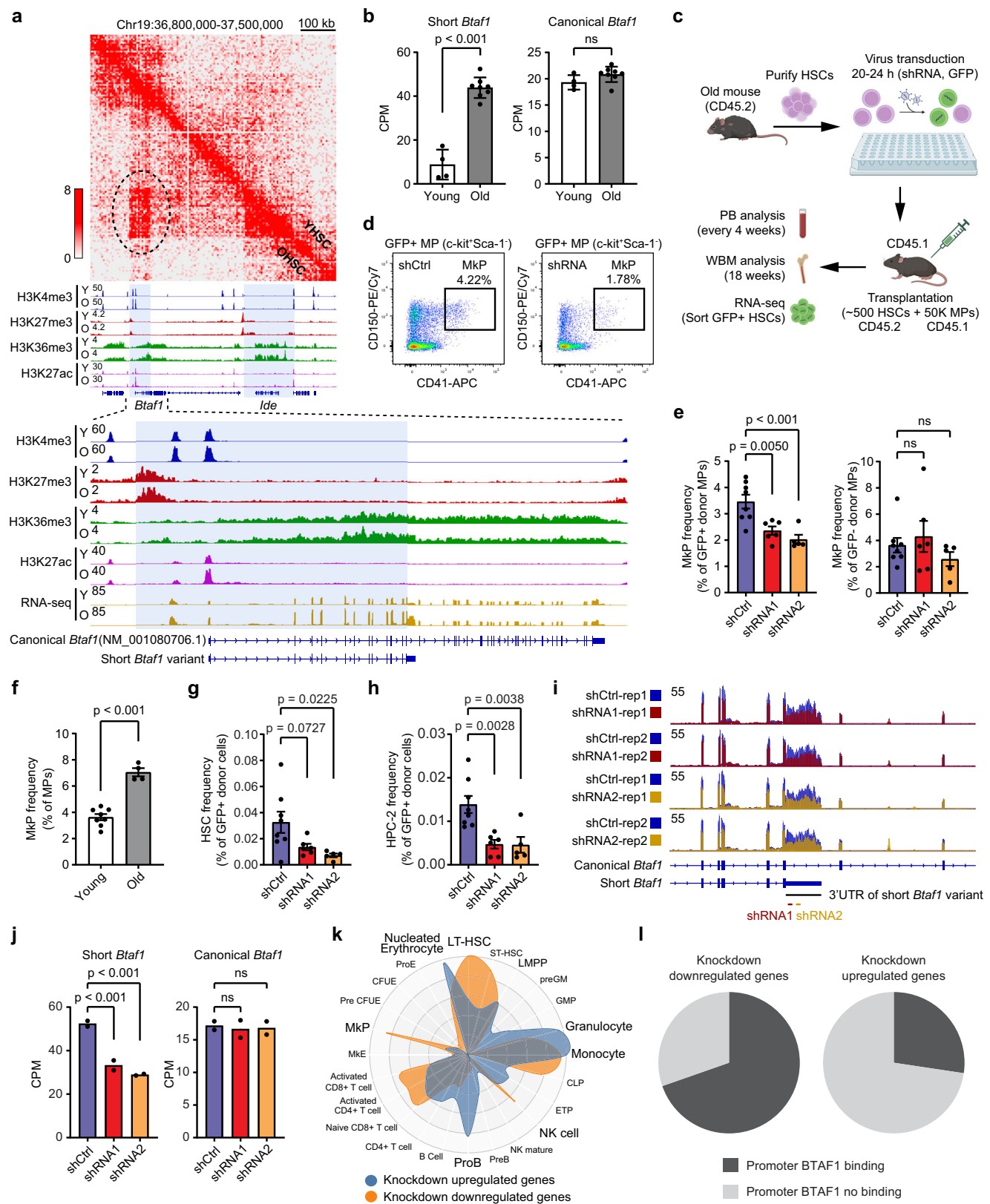

MEPs, and ChIP-seq showed that active histone marks at the *nBtaf1* locus were most enriched in MEPs (Supplementary Fig. 9). As MEPs give rise to MkPs, these findings support a role for *nBtaf1* in regulating MkP differentiation.

To address how the knockdown of *nBtaf1* affects global transcription in the stem cells, we performed RNA-seq on purified GFP⁺ HSCs isolated from recipient mice. Quantification of the shorter and

canonical *Btaf1* variant in these GFP⁺ HSCs confirmed knockdown of *nBtaf1* without affecting the expression of the canonical transcript (Fig. 4i, j, and Supplementary Fig. 7g). DEG analysis identified 95 upregulated and 56 downregulated genes in *nBtaf1* knockdown HSCs (Supplementary Fig. 10a). CellRadar (https://karlssong.github.io/cellradar/) analysis showed the downregulated genes after knockdown were enriched in LT-HSC and MkP (Fig. 4k), consistent with the

**Fig. 4 | Elevated expression of a short *Btaf1* variant in OHSCs contributes to HSC expansion and MkP differentiation. a** Hi-C heatmaps showed a new loop formed between part of *Btaf1* and the whole *Ide* in OHSCs. **b** Quantification of the shorter and canonical *Btaf1* variants in young (*n* = 4) and old (*n* = 8) HSCs. Mean ± SD, Wald test (two-sided). ns, not significant. **c** Experiment workflow of shRNA-mediated knockdown targeting the shorter *Btaf1* variant in OHSCs followed by transplantation. Created in BioRender. Ma, F. (2026) https://BioRender.com/a4qwrj6. **d** FACS plots show examples of MkP gating and frequency. **e** MkP frequency in donor-derived GFP+ and GFP- myeloid progenitors (MP). shCtrl (*n* = 8), shRNA1 (*n* = 6), shRNA2 (*n* = 5). Mean ± SEM, ordinary one-way ANOVA. Source data are provided as a Source data file. **f** MkP frequency in young and old mice. Young (*n* = 8), old (*n* = 4). Mean ± SEM, two-tailed t-test. Source data are provided as a Source data file. HSC (**g**) and HPC-2 (**h**) frequency in control and knockdown donor cells. shCtrl (*n* = 8),

shRNA1 (*n* = 6), shRNA2 (*n* = 5). Mean ± SEM, ordinary one-way ANOVA. Source data are provided as a Source data file. **i** Gene expression tracks around the 3′UTR of the shorter *Btaf1* variant. RNA-seq data of GFP-positive HSCs (LSKCD48⁻CD150⁺) purified from transplanted mice were used (*n* = 2 pooled samples of 2–4 recipient animals). **j** Quantification of the shorter and canonical *Btaf1* variants in control and knockdown HSCs from transplanted mice. shCtrl (*n* = 2), shRNA1 (*n* = 2), shRNA2 (*n* = 2). Mean ± SD, *n* = 2, Wald test (two-sided). **k** CellRadar plot derived from DEGs of control and knockdown comparison (FC > 1.5, *p* < 0.05). Wald test (two-sided) with Benjamini–Hochberg adjustment for multiple comparisons. Control HSCs, *n* = 2; knockdown HSCs, *n* = 4. **l** BTAF1 binding status at up- and down-regulated genes after knockdown of the shorter *Btaf1* variant. BTAF1 ChIP-seq data of mouse ESCs was used[50].

transplantation data showing reduced frequency of HSC and MkP from *nBtaf1* knockdown OHSCs (Fig. 4e, g). To examine whether BTAF1 directly regulates these identified DEGs, we analyzed published BTAF1 ChIP-seq data from mouse ESCs[50]. There was a clear enrichment for direct BTAF1 targets in the downregulated DEGs (70%) compared to upregulated DEGs (27%) (Fig. 4l). These data support an important role of *nBtaf1* in regulating age-associated HSC expansion and MkP differentiation.

### Knockdown of the shorter *Btaf1* variant reverses the aged HSC transcriptome

We then examined the impact of decreased *nBtaf1* expression on the HSC aging transcriptome. Previous literature had defined HSC aging signature (AS) genes characterized by being differentially expressed in at least 4 independent studies[13]. Using these AS gene lists, we performed gene set enrichment analysis (GSEA)[51,52] with transcriptome data from *nBtaf1* knockdown and control OHSCs. The results showed control OHSCs were enriched for expression of the AS genes that were upregulated in OHSCs (Fig. 5a, upper left panel), while *nBtaf1* knockdown HSCs were enriched for AS genes downregulated in OHSCs—or more robustly expressed in YHSCs (Fig. 5a, upper right panel). This indicates knockdown of *nBtaf1* shifts the aging transcriptome towards a younger state. Since the number of AS genes is small, and most of them overlapped with DEGs identified in our young versus old HSC RNA-seq comparisons (Supplementary Fig. 10b, c), we applied GSEA analysis with the DEGs we identified in this study. Consistent with the analysis using AS genes, the analysis with our DEGs showed similar results: *nBtaf1* knockdown HSCs reversed the expression pattern of DEGs found in OHSCs (*nBtaf1* knockdown increased expression of the age-associated downregulated transcripts and decreased expression of the upregulated transcripts) (Fig. 5a, lower panel). Direct comparison of the expression levels of AS genes and DEGs identified in this study showed the upregulated genes in OHSCs were significantly downregulated in *nBtaf1* knockdown HSCs, and the downregulated genes in OHSCs were significantly upregulated in *nBtaf1* knockdown HSCs (Fig. 5b). PCA analysis showed *nBtaf1* knockdown shifted the global transcriptome towards the young HSC state along PC1 (Supplementary Fig. 10d). CellRadar plots generated with the upregulated genes in OHSCs and downregulated genes in *nBtaf1* knockdown HSCs highly overlapped (Fig. 5c), and similar results were obtained with the downregulated genes in OHSCs and upregulated genes in *nBtaf1* knockdown HSCs (Fig. 5d). Together, these data support the knockdown of the shorter *Btaf1* variant reverses the HSC aging transcriptome.

The canonical isoform of BTAF1 is a TAF that can bind TATA-binding protein (TBP) to form the B-TFIID complex and regulate Pol II transcription[46,53]. It has an N-terminal TBP binding domain and an ATPase domain at the C-terminus (Fig. 5e). The shorter BTAF1 isoform retains the N-terminal TBP binding domain but lacks the C-terminal ATPase domain (Fig. 5e). It has been shown that BTAF1 acts as a repressor of Pol II transcription by dissociating TBP from TATA DNA

using the energy of ATP hydrolysis[53]. Our data suggest BTAF1 binds to the promoter of HSC and MkP-related genes (Fig. 4k, l). According to these data, we propose a model that illustrates how the shorter BTAF1 isoform drives the HSC aging transcriptome (Fig. 5f). In young or old HSCs, TBP binds to the promoter of HSC and MkP-related genes and recruits Pol II to start transcription. In young HSCs, the canonical BTAF1 isoform could bind to TBP and dissociate it from the promoter using the energy of ATP hydrolysis to repress transcription. The binding and removal of TBP to the promoter is reversible, which helps maintain the gene expression at appropriate levels in young HSCs. In old HSCs, high levels of the shorter BTAF1 isoform outcompetes the canonical BTAF1 isoform for TBP binding. Since the BTAF1 isoform lacks the ATPase domain, it cannot remove TBP from DNA, leading to persistent expression of its target genes and increased HSC self-renewal and MkP differentiation.

## Discussion

Our epigenetic analysis of HSCs during aging, starting with chromatin accessibility analysis using ATAC-seq, showed the chromatin in OHSCs is more open compared to young HSC, similar to results from other published chromatin accessibility studies[13,26,54]. Through our ChIP-seq analysis, we observed a global decrease in most histone modifications, including H3K4me3 and H3K27me3 at gene promoters, and H3K36me3 in gene bodies, we did not observe a global decrease in H3K27ac. The increased accessibility and maintenance of H3K27ac global levels support the IDEAS chromatin state analysis prediction that enhancer activity is increased in OHSCs.

H3K36me3 is positively correlated with active transcription[55], loss of H3K36me3 in OHSCs implies reduced transcription intensity, which is consistent with a study showing decreased transcription rate in OHSCs[56]. However, despite the reduced transcription rate, our data support the idea that the chromatin in OHSCs is more accessible for transcription machinery[13]. We found there are more regions with increased accessibility in OHSCs, and most of them are located at non-promoter regions and correlate with elevated TE expression and enhancer activity. As H3K36me3 and DNA methylation at gene bodies suppress cryptic transcription initiation[57], the globally decreased H3K36me3 in OHSCs also suggests elevated cryptic transcription, which was reported at the RNA level for OHSCs by the Dang lab[58].

We observed a similar number of bivalent promoters in young and old HSCs, and a trend of loss of bivalency in OHSCs. More specifically, most of the upregulated expression associated with bivalency changes featured loss of H3K27me3 at promoters, while the downregulated expression was associated with decreased H3K4me3, increased H3K27me3, or both. Of note, the promoter bivalency analysis in this study was done with bulk ChIP-seq, which cannot measure the state of promoter bivalency in individual cells. Future single-cell/single-molecule experiments will help resolve this current limitation of bulk ChIP-seq.

Previous studies had reported that OHSCs exhibit changes in nuclear volume and shape, proximity of chromosome 11 homologs, and attrition of X chromosome inactivation[44,54]. Our detailed analysis

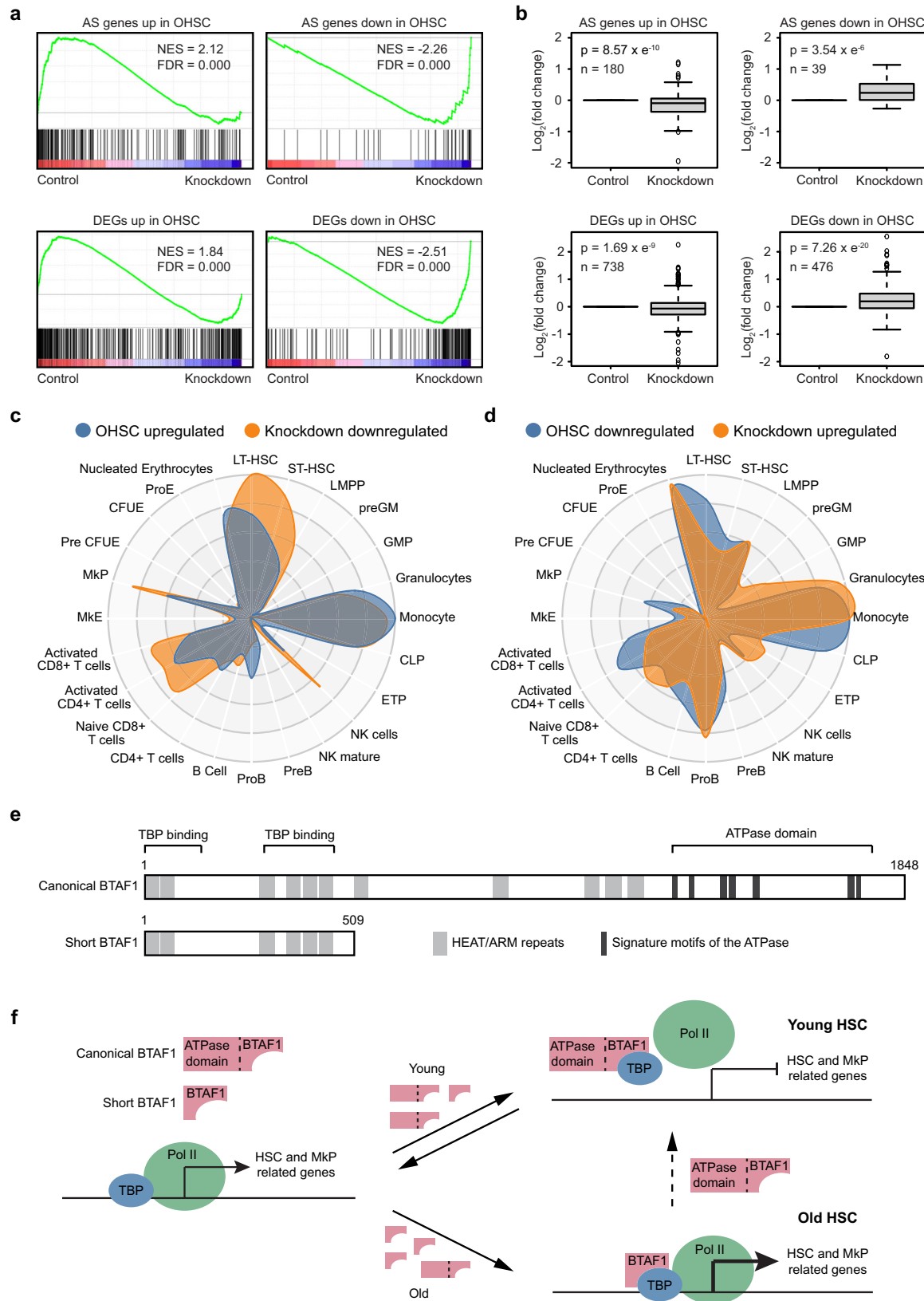

**e**

TBP binding    TBP binding                                    ATPase domain

Canonical BTAF1 ... 1 ... 1848

Short BTAF1 ... 1 ... 509

HEAT/ARM repeats    Signature motifs of the ATPase

**f**

Canonical BTAF1 — ATPase domain / BTAF1

Short BTAF1 — BTAF1

Young HSC — HSC and MkP related genes

Old HSC — HSC and MkP related genes

of chromatin architecture in young and old HSCs showed there are profound chromatin organization changes across all the chromosomes during aging. More specifically, we observed decreased long-range and increased short-range chromatin interactions in OHSCs. Interestingly, both types of changes primarily involved chromatin regions with heterochromatin features, including H3K9me3 and

H3K27me3. The decrease in long-range chromatin interactions was attributed largely to B–B compartment interactions, which may relate to the reduced expression of *Lmna* in OHSCs, as its encoding protein lamin A/C has been shown to be involved in regulating A/B compartment organization[43] and chromatin architecture changes during HSC aging[44]. Notably, our recent work with young and old pro-B cells

**Fig. 5 | Knockdown of the shorter *Btaf1* variant reverses the HSC aging transcriptome. a** GSEA plot generated with aging signature (AS) genes (upper panel) or DEGs from young old HSC comparison of this study (FC > 1.5, FDR < 0.05) (lower panel). RNA-seq data of HSCs purified from transplanted mice were used. Control HSCs, n = 2; knockdown HSCs, n = 4. **b** Box plots show gene expression changes of AS genes (upper panel) or DEGs from the young/old HSC comparison of this study (lower panel). Two-tailed t-test. Box plots show the median (center line), the 25th and 75th percentiles (box), and the 5th and 95th percentiles (whiskers). **c** CellRadar plot derived from upregulated DEGs in OHSCs and downregulated DEGs in knockdown HSCs. **d** CellRadar plot derived from downregulated DEGs in OHSCs and upregulated DEGs in knockdown HSCs. **e** Domain organization of the canonical

and shorter BTAF1 isoforms[46]. The locations of the TBP binding domain and ATPase domain were indicated. Gray boxes represent HEAT/ARM repeats. Black boxes indicate signature motifs of the ATPase. **f** Model illustrates how the shorter BTAF1 isoform drives the HSC aging transcriptome. In young HSCs, the canonical BTAF1 could bind to TBP at the promoter of HSC and MkP-related genes and dissociate it from DNA using the energy of ATP hydrolysis. This process is reversible, TBP can rebind to these promoters and restart transcription, helping to maintain gene expression at appropriate levels in young HSCs. In old HSCs, high levels of the shorter BTAF1 isoform outcompete the canonical BTAF1 isoform for TBP binding. As the shorter BTAF1 isoform lacks the ATPase domain, it cannot remove TBP from DNA, leading to consistent expression of its target genes.

showed age-associated increase of long-range and decrease of short-range chromatin interactions, which is opposite to the findings in OHSCs[33], indicating that different cell types may undergo varying changes in chromatin organization during aging.

Using the composite epigenetic profiles, we were able to identify a mRNA variant of *Btaf1* and provide evidence that its elevated levels in OHSCs drive the enhanced expression of HSC- and MkP-related genes associated with aging. In turn, this leads to HSC expansion and elevated production of MkPs in aged mice. As a key regulator of TBP binding and Pol II transcription, *Btaf1* is highly conserved between mouse and human[46]. By analyzing the long-read nanopore sequencing data of the human immune cells[59], we found human cells also express shortened *BTAF1* variants that lack the sequence for the C-terminal ATPase domain (Supplementary Fig. 10e), indicating the expression of shortened *BTAF1* variant may also be involved in regulating aging phenotypes in human cells.

Notably, although histone ChIP-seq with limited number of HSCs was successfully performed in this study, we systematically attempted TF ChIP-seq using multiple antibodies previously validated in bulk experiments (including Pol II, CTCF, RUNX1, EZH2, GATA2, among others). With the exception of PU.1, these efforts did not produce robust signals. This likely reflects fundamental differences between histone modifications and TFs in low-input ChIP-seq settings, such as the transient nature of TF–DNA interactions and limited antigen availability in quiescent HSCs. Therefore, while the optimized protocol is robust for profiling histone modifications, it is not broadly applicable to TF profiling in primary HSCs, preventing direct in vivo assessment of nBTAF1 function within HSCs.

In summary, we have generated high-resolution epigenetic maps of HSCs that allowed us to identify loci with age-associated changes in chromatin, histone modifications, and ultimately drive expression of gene variants, including *nBtaf1*, that contribute to aspects of HSC aging phenotypes. We propose isoforms of *BATF1* may also serve a therapeutic target to mitigate aspects of blood aging in humans.

## Methods
### Animals
Young C57BL/6J (CD45.2, JAX #:000664) male mice (3–4 months) were purchased from The Jackson Laboratory, and old C57BL/6J male mice (24–26 months) were acquired from the NIA Aged Rodent Colony. Young female transplant recipient B6.SJL-Ptprca Pepcb/BoyJ (CD45.1, JAX #:002014) mice were obtained from The Jackson Laboratory. Animals were co-housed in a barrier SPF facility and euthanized by $CO_2$ administration followed by cervical dislocation. All animal experiments were performed under protocols approved by NIA Institutional Animal Care and Use Committees (469-TGB-2025).

### Antibodies
**Flow cytometry.** The following antibodies are from Biolegend: Biotin anti-TER119 (Cat# 116204, 1:100 dilution), Biotin anti-B220 (Cat# 103204, 1:100 dilution), Biotin anti-CD3 (Cat# 100244, 1:100 dilution), Biotin anti-Mac1 (Cat# 101204, 1:100 dilution), Biotin anti-Gr1 (Cat# 108404, 1:100 dilution), Biotin anti-IL7Rα (Cat# 135005, 1:100

dilution), PB anti-TER119 (Cat# 116232, 1:200 dilution), PB anti-B220 (Cat# 103227, 1:200 dilution), PB anti-Mac1 (Cat# 101224, 1:200 dilution), PB anti-CD3 (Cat# 100214, 1:200 dilution), BV421 anti-Gr1 (Cat# 108445, 1:200 dilution), BV421 anti-IL7Rα (Cat# 135027, 1:200 dilution), APC/Cy7 anti-Sca1 (Cat# 108126, 1:200 dilution), PE anti-cKit (Cat# 105808, 1:200 dilution), APC anti-Flk2 (Cat# 135310, 1:50 dilution), PE/Cy7 anti-CD150 (Cat# 115914, 1:200 dilution), APC anti-CD48 (Cat# 103412, 1:200 dilution), APC anti-CD45.1 (Cat# 110714, 1:200 dilution), PB anti-CD45.2 (Cat# 109820, 1:100 dilution), PerCP/Cy5.5 anti-TER119 (Cat# 116228, 1:200 dilution), APC/Cy7 anti-CD45R/B220 (Cat# 103224, 1:200 dilution), PE anti-CD3 (Cat# 100206, 1:200 dilution), PE/Cy7 anti- Mac1 (Cat# 101216, 1:200 dilution), BV510 anti-Gr1 (Cat# 108457, 1:200 dilution), APC anti-CD41 (Cat# 133914, 1:200 dilution), PerCP/Cy5.5 anti-CD45.1 (Cat# 110728, 1:100 dilution). FITC anti-CD34 is from ThermoFischer (Cat# 11-0341-85, 1:50 dilution).

**ChIP-seq.** Anti-H3K4me3 (Sigma, Cat# 07-473), anti-H3K9me3 (abcam, Cat# ab8898), anti-H3K27me3 (Diagenode, Cat# C15410195), anti-H3K36me3 (abcam, Cat# ab9050), anti-H3K27ac (abcam, Cat# ab4729).

### HSC purification
BM cells were collected from crushed femurs, tibias, pelvises, humeri and filtered through 100 μm filters. Isolated BM cells were positively selected with PE c-kit antibody and EasySep™ PE Positive Selection Kit II (17684, STEMCELL). For wild-type (WT) mice, enriched cells were stained with antibodies against lineage (Ter119, B220, Mac1, CD3, Gr1, IL-7Rα), Sca-1, c-kit, CD34, Flk2, and CD150. For post-transplant recipient mice, enriched cells were stained with antibodies against lineage (Ter119, B220, Mac1, CD3, Gr1, IL-7Rα), Sca-1, c-kit, CD48, CD150, CD45.2. Propidium iodide (PI) was used to exclude dead cells. HSCs were sorted as PI⁻Lin⁻Sca-1⁺c-kit⁺(LSK)CD34⁻Flk2⁻CD150⁺ for WT mice and PI⁻LSKCD48⁻CD150⁺CD45.2⁺ for post-transplant recipient mice with BD FACSAria II Cell Sorter.

### HSC in vitro culture and virus transduction
HSCs were cultured in media described before[60]: F12 medium, 10 mM HEPES, 1% ITSX, 1% P/S/G, 100 ng/ml mouse TPO, 10 ng/ml mouse SCF and 0.1% PVA (P8136, Sigma) in 96-well U-bottom plates at 37 °C with 5% $CO_2$ and $O_2$. The lentivirus shRNA knockdown vector construction and virus packaging were performed by VectorBuilder. The target sequences of shRNA are 5′-TTCCTGGAGCCTCTCATAATT-3′ (shRNA1) and 5′-TTTCAGTTCTTGTCCTTAATT-3′ (shRNA2). For virus transduction, 10,000 purified OHSCs were cultured in 50 μl of media with the virus overnight (12–16 h). The next morning, 150 μl of fresh media was added to each well. For the knockdown efficiency test, HSCs were cultured for 3.5 days before sorting for RNA-seq. For transplantation, HSCs were cultured for 20–24 h before the procedure.

### HSC transplantation
About 500 cultured HSCs (CD45.2) were transplanted into lethally irradiated (9.56 Gy) recipients (CD45.1) together with 50,000 myeloid progenitor cells (CD45.1) via retro-orbital injection.

## Peripheral blood analysis

Peripheral blood analysis was performed at 4-week intervals post-transplantation. Population frequencies were determined using flow cytometry after ACK treatment and staining with CD45.1, CD45.2, Ter119, B220, Mac1, CD3, and Gr1.

## Whole bone marrow analysis

BM cells were treated with ACK and then stained with antibodies. For staining MkP in WT mice: lineage (Ter119, B220, Mac1, CD3, and Gr1), Sca-1, c-kit, CD41, and CD150. For staining MkP in post-transplant recipient mice: lineage (Ter119, B220, Mac1, CD3, and Gr1), Sca-1, c-kit, CD41, CD150, and CD45.2. For staining progenitor cells in post-transplant recipient mice: lineage (Ter119, B220, Mac1, CD3, and Gr1), Sca-1, c-kit, CD48, CD150, CD45.1, and CD45.2.

## In situ Hi-C

In situ Hi-C was performed as described previously[61] with modifications. 20,000 sorted HSCs were crosslinked with 2% formaldehyde at RT for 10 min with rotation. Crosslinking was stopped by adding 2 M glycine to a final concentration of 0.2 mM and incubate at RT for 5 min. After one wash with cold 1× PBS, pelleted cells were stored at −80 °C. Cells were thawed on ice, and 160 μl of cold lysis buffer (10 mM Tris-HCl, 10 mM NaCl, 0.2% IGEPAL CA-630, 1× proteinase inhibitor cocktail in water) was added. Cell suspensions were incubated on ice for 20 min and then centrifuged at 3000 × g for 8 min at 4 °C. Pellets were resuspended in 25 μl of 0.5% SDS and incubated at 62 °C for 10 min. 73 μl of water and 12.5 μl of 10% Triton X-100 were added to quench SDS and incubated at 37 °C for 15 min. 12.5 μl of 10× NEBuffer 2 and 2 μl of MboI (R0147M, NEB) were added, and chromatin was digested at 37 °C overnight in a thermomixer with 1000 rpm mixing. Tubes were incubated at 62 °C for 20 min to inactivate MboI and cooled to room temperature (RT). 25 μl of fill-in master mix containing 18.75 μl of 0.4 mM biotin-14-dATP (19524016, ThermoFisher), 0.75 μl of 10 mM dTTP (18255018, ThermoFisher), 0.75 μl of 10 mM dCTP (18253013, ThermoFisher), 0.75 μl of 10 mM dGTP (18254011, ThermoFisher), 4 μl of DNA Polymerase I Large (Klenow) Fragment (M0210L, NEB) was added, and incubated in a thermomixer at 37 °C for 1.5 h with 1,000 rpm mixing. 450 μl of ligation master mix containing 60 μl of T4 DNA ligase buffer (B0202S, NEB), 47.5 μl of 10% Triton X-100, 3 μl of 20 mg/ml BSA (B9000S, NEB), 5 μl of 400 U/μl T4 DNA Ligase (M0202L, NEB), 334.5 μl of water was next added and incubated at RT for 4 h with slow rotation. Tubes were centrifuged at 3000 × g for 8 min at 4 °C, and pellets resuspended with 140 μl of SDS lysis buffer (17–295, Sigma-Aldrich) and incubated on ice for 10 min. Bioruptor Plus (Diagnode) sonication was performed with the following setting: 9 cycles of 30 s ON and 30 s OFF at LOW setting. 5.85 μl of 5 M NaCl was added and incubated overnight at 65 °C with 1,000 rpm mixing. 6.1 μl of 20 mg/ml proteinase K (26160, ThermoFisher) was then added and incubated at 55 °C for 2 h. DNA was purified with 1.6x SPRIselect beads (B23318, Beckman Coulter). Libraries were prepared with TruSeq Nano DNA Low Throughput Library Prep Kit (20015964, Illumina) according to the manufacturer's instructions. Pull-down of biotin-labeled DNA was performed as described previously[61]. Sequencing was done on the Illumina HiSeq 2500 instrument.

## ATAC-seq

ATAC-seq was performed as described previously[62]. Six thousand sorted HSCs were spun down using a fixed angle rotor at 500 g, 4 °C for 10 min. Cells were resuspended with 50 μl of cold resuspension buffer (10 mM Tris-HCl, 10 mM NaCl, 3 mM MgCl$_2$ in water) containing 0.1% NP40, 0.1% Tween-20, and 0.01% Digitonin by pipetting up and down 3 times, and incubated on ice for 3 min. One milliliter of cold resuspension buffer containing 0.1% Tween-20 (without NP40 or Digitonin) was added to wash the cells, and tubes were inverted 3 times to mix. The nuclei were pelleted by spinning at 500 g, 4 °C for 10 min with a fixed-angle rotor. The pellet was then resuspended in 50 μl of transposition mix (25 μl of 2x TD buffer, 2.5 μl of Tn5 Transposase (20034197, Illumina), 22.5 μl of H2O) by pipetting up and down 6 times. The reaction was incubated at 37 °C for 30 min in a thermomixer with 1,000 rpm mixing, and cleanup was performed with DNA Clean & Concentrator-5 (D4013, Zymo Research). DNA was eluted with 21 μl of H2O and amplified for 5 cycles with NEBNext Ultra II Q5 Master Mix (M0544L, NEB). We determined the optimum additional PCR cycles by qPCR as described previously[63]. The final PCR reaction was purified with DNA Clean & Concentrator-5 (D4013, Zymo Research). Libraries were size-selected (120–800 bp) by running TBE gel (EC6265BOX, ThermoFisher) and sequenced on an Illumina HiSeq 2500 instrument.

## ChIP-seq

ChIP-seq was performed as described previously[27] with Chromatin Immunoprecipitation (ChIP) Assay Kit (17–295, Millipore). Ten thousand sorted HSCs were mixed with $1 × 10^8$ bacteria (18258012, ThermoFisher), then crosslinked with 1% formaldehyde (F8775, Sigma-Aldrich) at RT for 10 min. Crosslinking was stopped by adding 2 M glycine to a final concentration of 0.125 mM and incubated at RT for 5 min. After one wash with cold 1× PBS, cells were resuspended in SDS lysis buffer with 1× proteinase inhibitor cocktail (Millipore, 4693132001) and incubated on ice for 10 min. Sonication was performed with Bioruptor Plus (Diagnode) using the following setting: 3 × 10 cycles of 30 s ON and 30 s OFF at HIGH setting. After sonication, fragmented chromatin was diluted by adding 9 volumes of ChIP dilution buffer with 1× proteinase inhibitor cocktail. Corresponding antibodies were added and incubated for 12 h at 4 °C with rotation. Twenty microliters of Dynabeads Protein A (10002D, ThermoFisher) were added and incubated at 4 °C for 2 h with rotation. Before washing, 5 ng biotin-DNA carrier immobilized with 10 μl of Dynabeads M-280 Streptavidin (11205D, ThermoFisher) was added. Sample was washed with 1 ml washing buffer at 4 °C for 5 min in the following order: 1× with low salt buffer, 1× with high salt buffer, 1× with LiCl washing buffer, 2× with TE. Beads were resuspended with decrosslinking buffer (0.5% SDS, 0.2 M NaCl in TE) and incubated at 65 °C overnight with 1000 rpm mixing. 20 mg/ml proteinase K (final concentration 0.8 mg/ml, 26160, ThermoFisher) was added and incubated at 55 °C for 2 h. ChIPed DNA was purified with 1.6x SPRIselect beads (B23318, Beckman Coulter). Libraries were prepared with TruSeq Nano DNA Low Throughput Library Prep Kit (20015964, Illumina) according to manufacturer's instructions. Sequencing was done with the Illumina HiSeq 2500 instrument.

## RNA-seq

HSCs were directly sorted into TRIzol Reagent (15596026, Thermo-Fisher) from ether-pooled young or individual old mice and stored at −80 °C. RNA was purified with Direct-zol RNA Microprep (R2060, Zymo) for WT young and old HSCs. RNA from transplanted HSCs was extracted using phenol-chloroform extraction and isopropanol precipitation. cDNA libraries were prepared with SMART-Seq RNA Kit (634891 for RNA from young and old HSCs, 634772 for RNA from transplanted HSCs, TaKaRa) according to the manufacturer's protocol. Sequencing libraries were constructed using Nextera XT DNA Library Preparation Kit (FC-131-1024, Illumina) with 125 pg input cDNA. Sequencing was done with the Illumina HiSeq 2500 instrument for WT young and old HSC libraries, and the Illumina NovaSeq 6000 System for libraries from transplanted HSCs.

## Hi-C analysis

Raw Hi-C reads in FASTQ files were aligned to the reference genome mm10 by HiCUP (v 0.7.2)[64]. High-quality unique paired-end tags (PET) (MAPQ > = 10 from HiCUP) were further converted to HIC files by Juicer (v1.6.0)[65] for visualization and cLoops2 (v0.0.3)[66] data directories for quantification, similarity, and aggregation analysis. Hi-C compartment

eigenvectors were calculated by hicPCA from HiCExplorer (v3.6)[67] at the resolution of 100 kb, and TADs were called by Juicer at the resolution of 25 kb.

## ATAC-seq analysis

All sequencing reads were trimmed using cutadapt[68], and trimmed reads (>36 bp minimum alignment length) were mapped against the mm10 genome using the BWA aligner[69]. We used de-duplicated and uniquely mapped reads for peak calling analysis after excluding high-sensitive black-list regions defined by ENCODE. The candidate peaks were predicted by MACS peak calling software (FDR < 0.05)[70]. After identifying narrow peaks from young and old HSC replicates, we created a merged set of consensus peaks and generated a matrix of open chromatin regions (OCR). This OCR matrix was then imported into the R package DESeq2[71], and we determined DARs with cutoff: FC > 1.5, CPM > 1.5, FDR < 0.01. Finally, the candidate differential OCRs were submitted to search for potential TF binding sites using HOMER software[72] with non-DARs as background regions. In this analysis, de novo motif and known motif searches were performed, and we reported the top five significant de novo motif results. Composite plots and heatmaps were generated by customized Python script and Java TreeView software.

## ChIP-seq analysis

All sequencing reads were trimmed using cutadapt[68], and trimmed reads (>36 bp minimum alignment length) were mapped against the mm10 genome using the BWA aligner[69]. We used de-duplicated and uniquely mapped reads for peak calling analysis after excluding high-sensitive black-list regions defined by ENCODE. For H3K4me3 and H3K27ac histone marks, the narrow candidate peaks were predicted by MACS peak calling software (FDR < 0.05)[70]. The sequencing libraries of all the histone marks were normalized (10 M reads), and we summarized the average peak size to 1 kb binning across all mouse genomes. These normalized files were used to perform broad peak calling and chromatin state analysis (e.g., IDEAS[36]). For H3K27me3, we identified broad peaks from IDEAS chromatin states (FDR < 0.05) and expanded the peak range to include signals that appeared significant compared to random background height (average broad peak height > 0.3, and significant $p < 1e^{-4}$). To identify histone peaks that changed between age groups, we used cutoff FC > 1.5 (merged data), and each replicate from one age group should have an FC of at least 1.2 compared to either replicate from the other age group. To analyze bivalent promoters, we first checked whether an H3K4me3 or H3K27me3 peak could be assigned to a gene promoter. For H3K4me3 peaks, if the peak was within 1 kb of the TSS of a gene, it was assigned to this gene. For H3K27me3 peaks, if the peak overlapped with TSS or within 2 kb downstream of the TSS of a gene, it was assigned to this gene. A gene is defined in this study as having a bivalent promoter if there are both H3K4me3 and H3K27me3 peaks assigned to this gene. Composite plots and heatmaps were generated with customized Python script and Java TreeView software.

## RNA-seq analysis

For analysis of transcriptome datasets, we built an index for STAR using the GENCODE M22 reference feature, including protein-coding and noncoding genes. Prior to sequence alignment, we applied trim galore (version 0.4.3) with cutadapt (version 1.12)[68] to remove any unnecessary genomic fragments (e.g., adapter dimers) and low-quality nucleotide sequences from the raw reads. We mapped adapter-trimmed sequencing reads to the mouse reference genome (mm10) using the STAR aligner[73] and calculated the raw count using feature-Counts software (gene-level)[74]. DEG lists were generated with DESeq2[71] using the cutoff: FC > 1.5, FDR < 0.05. For transposable element detection, we utilized SQuIRE[75] and used limma based edgeR package of R[76] to find differential transposable elements.

## Reporting summary

Further information on research design is available in the Nature Portfolio Reporting Summary linked to this article.

## Data availability

The raw data generated in this study have been deposited in the NCBI Gene Expression Omnibus database under accession code GSE204933. The other sequencing data used in this study are available in the NCBI Gene Expression Omnibus database under accession code GSE284988. These data can also be explored through an online browser at: https://genomicspark.shinyapps.io/Zong_Park_Epigenetic_Aging/. Source data are provided with this paper.

## Code availability

All customized Python scripts used in the manuscript are available via GitHub repository URL (https://github.com/genomicspark/ESCA_Unit_Scripts) and has been archived in Zenodo for citation (https://doi.org/10.5281/zenodo.17857712)[77].

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

## Acknowledgements

Many thanks to Drs. Rafael de Cabo, Michel Bernier, Ranjan Sen, Hagai Yanai and all members of TGB for invaluable support. We would also like to thank Drs. Emmanouil Maragkakis and Cedric Belair for their suggestions on nanopore data analysis. Kind support was shared by the Genomics Core, and we'd like to specially acknowledge William Wood, Jinshui Fan, and Supriyo De for their assistance with data handling and sequencing advice. Thanks to Christopher Dunn at the NIA Flow Core for always being helpful and willing to share time and expertise. We would like to thank all the members of the NIA Comparative Medicine Section for their consistent efforts and high standards of animal care. Data analysis of this work utilized the computational resources of the NIH HPC Biowulf cluster (http://hpc.nih.gov). This research was supported entirely by the Intramural Research Program of the NIH, National Institute on Aging.

## Author contributions

L.Z. designed and conducted experiments, analyzed the data, and wrote the manuscript; B.P. performed bioinformatical analyses and edited the manuscript; Y.C., F.M. critically analyzed the Hi-C data; F.T., W.K. helped with the transplantation experiment. K.Z. participated in study design; I.B. supervised the study and edited the manuscript.

## Competing interests

The authors declare no competing interests.

## Additional information

**Supplementary information** The online version contains Supplementary material available at https://doi.org/10.1038/s41467-026-70787-4.

