## [Transparent Peer Review File · Nature Communications]

Chromatin reorganization drives overexpression of a Btaf1 variant underpinning hematopoietic aging

Corresponding Author: Dr Isabel Beerman

Version 0:

Reviewer comments:

Reviewer #2

(Remarks to the Author)

The authors have answered all my comments

Reviewer #3

(Remarks to the Author)

As I mentioned in my initial review, this is a well performed study that is data-heavy and could constitute a nice resource for the broader community. At the same time, I still find the study very descriptive in its bulk and the characterization of the nBTAF1 variant rather superficial. Essentially, the authors claim that doing more functional work is hindered by the limited HSC numbers they get from each mouse, while at the same time having managed to perform Hi-C and ChIP on those same cells. Therefore, it must certainly be doable to perform some of the proposed nBTAF1 experiments that would explain how this isoform actually works. In my view, an RNA-seq experiment alone is not enough and only shows some gene expression 'rescue' patterns. Moreover, the Hi-C data presented is still mostly low resolution and not really used to infer regulation (there is a single nice example in Fig. 4 and nothing else at sub-100-kbp resolution), including that by nBTAF1. If these issues are not resolved, I cannot support publication of the manuscript as it stands.

Reviewer #4

(Remarks to the Author)

The authors did a very significant effort to respond to Reviewer 1. The responses addressed the concerns of the reviewer

Version 1:

Reviewer comments:

Reviewer #3

(Remarks to the Author)

I want to thank the authors for the detailed explanation on the limitations of their study system. In fact, it would be nice if (some of) these also appeared in the manuscript text for readers to fully grasp why some things were done and others not. Nonetheless, I think the manuscript should be accepted for publication in Nat Comms.

Reviewer #3 (Remarks to the Author):

As I mentioned in my initial review, this is a well performed study that is data-heavy and could constitute a nice resource for the broader community. At the same time, I still find the study very descriptive in its bulk and the characterization of the nBTAF1 variant rather superficial. Essentially, the authors claim that doing more functional work is hindered by the limited HSC numbers they get from each mouse, while at the same time having managed to perform Hi-C and ChIP on those same cells. Therefore, it must certainly be doable to perform some of the proposed nBTAF1 experiments that would explain how this isoform actually works. In my view, an RNA-seq experiment alone is not enough and only shows some gene expression 'rescue' patterns. Moreover, the Hi-C data presented is still mostly low resolution and not really used to infer regulation (there is a single nice example in Fig. 4 and nothing else at sub-100-kbp resolution), including that by nBTAF1. If these issues are not resolved, I cannot support publication of the manuscript as it stands.

Response:

We thank the reviewer for again acknowledging the technical quality of the work and the value of the dataset to the broader community. We also appreciate the reviewer's continued emphasis on strengthening the mechanistic understanding of the nBTAF1 variant. Below, we detail the extensive experimental efforts undertaken to address these concerns, the technical and biological limitations encountered, and why we believe the current data represent the most rigorous and informative analysis achievable in primary HSCs.

1. Limitations imposed by primary HSC biology and cell number

Primary murine HSCs are extremely rare (~3,000 HSCs per young mouse), and high-resolution 3D genome organization and epigenetic regulation in HSCs - particularly during aging - remain largely unknown. In this study, we optimized histone ChIP-seq protocols for low cell numbers (~10,000 HSCs), generating epigenetic maps with substantially higher resolution than previously published datasets. These data uncovered previously unrecognized epigenetic features associated with HSC aging.

While histone ChIP-seq was successful, we also systematically attempted transcription-factor ChIP-seq using multiple antibodies validated in bulk experiments (Pol II, CTCF, RUNX1, EZH2, GATA2, among others). With the exception of PU.1, these efforts did not yield successful signal. This likely reflects fundamental differences between histone modifications and transcription factors in low-input ChIP-seq, including transient DNA binding and limited antigen availability in quiescent HSCs. Thus, the optimized protocol is robust for histone modifications but not broadly applicable to transcription-factor profiling in primary HSCs. We additionally tested CUT&RUN approaches for transcription factors in HSCs. Although CUT&RUN can reduce input requirements, in our hands these experiments yielded poor resolution with 10K HSCs.

2. Attempts to functionally interrogate nBTAF1 in vivo

To directly test whether nBTAF1 overexpression could induce aging-like phenotypes, we attempted lentiviral overexpression of nBtaf1 in young HSCs followed by transplantation into young recipients. While efficient overexpression could be achieved *in vitro* (Fig. R1a), nBtaf1 expression was not stably maintained in transplanted HSCs *in vivo* after 19–20 weeks (Fig. R1b). This variability prevented meaningful downstream functional or epigenetic analyses in this setting.

Fig. R1a

Fig. R1b

3. Context dependence of nBTAF1 function

A critical observation from our study is the strong context dependence of nBTAF1 function. Primary HSCs *in vivo* are largely quiescent and embedded in a complex niche, whereas cultured HSCs are activated, prone to differentiation, and lack physiological cues. This distinction likely explains the failure to maintain nBtaf1 overexpression *in vivo*. Consistent with this, improved aging-associated transcriptomic changes following nBtaf1 knockdown were observed only in transplanted HSCs *in vivo*, but not in *in vitro*-cultured cells (Fig. R2).

Fig. R2

4. Lineage-specific evidence supporting a role for nBTAF1

Guided by our transplantation data showing that nBTAF1 promotes megakaryocyte progenitor (MKP) differentiation, we examined nBtaf1 expression and chromatin state across myeloid progenitors (CMP, GMP, and MEP). RNA-seq revealed the highest nBtaf1 expression in MEPs, and CHIP-seq showed that active histone marks (H3K4me3, H3K27ac, and H3K36me3) at the nBtaf1 locus were strongest in MEPs (Extended Data Fig. 9). As MEPs give rise to MkPs, these findings provide independent, lineage-level support for the functional conclusions drawn from our *in vivo* experiments.

Extended Data Fig. 9

5. Hi-C resolution and scope of regulatory inference

For Hi-C experiments, we started with 20,000 sorted HSCs per sample, which is substantially lower than typical bulk Hi-C input. Resolution estimation analyses indicated that the majority of paired-end tags (PETs) are captured at 25-kb resolution, with ~80% or more represented at 50-kb resolution (Extended Data Fig. 5c). These metrics demonstrate that our Hi-C data are well powered for analyses at ≥ 25 -kb resolution. Accordingly, the primary focus of our Hi-C analysis is on chromatin compartments and topologically associating domains (TADs) level, for which this resolution is appropriate and reliable. We agree that the dataset is not suitable for enhancer-promoter-level regulatory inference, and we therefore intentionally limited our analyses to the TAD level. The nBtaf1 example highlighted in Fig. 4 was identified through unbiased screening of TADs showing the most pronounced structural changes, rather than through targeted fine-scale regulatory modeling.

Extended Data Fig. 5c

Conclusion

In summary, we made extensive attempts to further dissect the mechanistic role of nBTAF1 using complementary approaches. The limitations encountered reflect fundamental constraints of primary HSC biology rather than a lack of experimental effort. We believe that the combination of high-resolution epigenomic profiling, rigorous transplantation assays, and targeted functional perturbations provides the strongest achievable evidence for the role of nBTAF1 in HSC aging and lineage bias, while also establishing a valuable resource for the field.